# Thermal Treatment to Obtain 5-Hydroxymethyl Furfural (5-HMF), Furfural and Phenolic Compounds from Vinasse Waste from *Agave*

**DOI:** 10.3390/molecules28031063

**Published:** 2023-01-20

**Authors:** Miguel Angel Lorenzo-Santiago, Jacobo Rodríguez-Campos, Rodolfo Rendón-Villalobos, Edgar García-Hernández, Alba Adriana Vallejo-Cardona, Silvia Maribel Contreras-Ramos

**Affiliations:** 1Unidad de Tecnología Ambiental, Centro de Investigación y Asistencia en Tecnología y Diseño del Estado de Jalisco A.C. (CIATEJ), Normalistas No. 800, Colinas de la Normal, Guadalajara C.P. 44270, Jalisco, Mexico; 2Unidad de Servicios Analíticos y Metrológicos (CIATEJ), Normalistas No. 800, Colinas de la Normal, Guadalajara C.P. 44270, Jalisco, Mexico; 3Centro de Desarrollo de Productos Bióticos, Instituto Politécnico Nacional (IPN), Calle Ceprobi número 8, Colonia San Isidro C.P. 62731, Morelos, Mexico; 4Tecnológico Nacional de México, I.T. Zacatepec, Calzada Tecnológico No.27, Colonia Centro, Zacatepec C.P. 62780, Morelos, Mexico; 5Unidad de Biotecnología Médica y Farmacéutica, Centro de Investigación y Asistencia en Tecnología y Diseño del Estado de Jalisco A.C. (CIATEJ), Normalistas No. 800, Colinas de la Normal, Guadalajara C.P. 44270, Jalisco, Mexico

**Keywords:** agro-industrial residues, phenolic compounds, furan derivatives, sugar transformation, tequila vinasse

## Abstract

Vinasses represent important final disposal problems due to their physical-chemical composition. This work analyzed the composition of tequila vinasses and increased 5-hydroxymethylfurfural, furfural, and phenolic compounds using thermal hydrolysis with hydrogen peroxide as a catalyst. A statistical Taguchi design was used, and a UPLC-MS (XEVO TQS Micro) analysis determined the presence and increase of the components. The treatment at 130 °C, 40 min, and 0.5% of catalyst presented the highest increase for 5-HMF (127 mg/L), furfural (3.07 mg/L), and phenol compounds as chlorogenic (0.36 mg/L), and vanillic acid (2.75 mg/L). Additionally, the highest removal of total sugars (57.3%), sucrose (99.3%), and COD (32.9%). For the treatment T130:30m:0P the syringic (0.74 mg/L) and coumaric (0.013 mg/L) acids obtained the highest increase, and the treatment T120:30m:1P increased 3-hydroxybenzoic (1.30 mg/L) and sinapic (0.06 mg/L) acid. The revaluation of vinasses through thermal treatments provides guidelines to reduce the impact generated on the environment.

## 1. Introduction

By 2021, ethanol production was 103 million cubic meters worldwide, headed by the United States and Brazil with 55 and 27%, respectively [1]. Sica et al. [2] reported, the ethanol production process produces between 10 and 15 L of vinasses per liter of processed ethanol. It is considered that by 2021 the worldwide production of vinasses was 1.2 × 10^9^ cubic meters. 

Vinasses are the liquid waste that results from the end of the distillation process of ethanol [3,4], with different sources such as sugarcane [5], orange [6], and agave [4], among others. Vinasses contain high concentrations of organic matter, low pH, electric conductivity, and mineral salts [2,5] and are considered the largest source of contamination in ethanol production or distilled industry [7].

Its main uses are in irrigation in agricultural soils or discharged directly into bodies of water [2,4]. Although they are used to irrigate and fertilize the field, they contain low macronutrients and micronutrients [8]. It generates changes in the structure and salinity of the soil, high harmfulness, and bioavailability of nutrients, being the aromatic metabolites (furans and phenols) are the main ones responsible for the toxicity of vinasses [9]. Additionally, the vinasses can threaten water resources, contribute to greenhouse gas generation, and disturb microbial decomposers’ communities [10].

Tequila is a distilled beverage that is recognized worldwide. Tequila is obtained from *Agave tequilana* Weber Va. Azul. This industry has severe environmental problems due to the final disposal of the waste generated (vinasses and bagasse) before and after production [9]. It is estimated that of the total processed agave, 40% is bagasse, and 1 L of tequila, produces between 7 and 10 L of vinasse [11]. According to the Tequila Regulatory Council (Consejo Regulador del Tequila, CRT), for 2021, more than 2000 ton of *Agave tequilana* Weber was used to produce 527 million liters of tequila, generating 800 ton de agave and 4.47 × 10^9^ L de vinasses, approximately [12].

Vinasses are composed mainly of water, fermentable sugars, organic acids, esters, and alcohols, present low pH, high organic matter content measured as chemical oxygen demand (COD), high concentrations of lipids and fats, Ca^+2^, Mg^+2^, K^+^, and phenols [4,13].

One of the main problems in recovering these components is the high costs of specific treatments, catalysts (acid, salts, or metals), and the type of raw material [14]. Currently, the alternatives for the recovery of furans and phenols are based on the use and revaluation of renewable biomass, agro-industrial wastes, edible biomasses, non-edible lignocellulosic biomasses, food wastes, and fruit residues (corn cob waste, kiwi pulp, mango residues, cane bagasse, wheat straw, watermelon peel, among others). It is a potential solution that helps reduce the impact that biomass and agro-industrial waste cause on the environment [15,16].

The production of furans and phenols from tequila vinasse has been proposed [13]. The polarities of furans and phenolic compounds vary significantly; the same happens with carbohydrates, making it difficult to recover these economically important compounds [17]. At present, high-strength cellulose resin [18], immobilized Enzymes, and Cation Exchange Resin [19] are the principal isolation methods for this compound. 

Hydrogen peroxide (H_2_O_2_) was selected as a green catalyst for its low cost and environmental impact after the treatments. It is a less aggressive compound than H_2_SO_4,_ a catalyst widely used in the chemical transformation of sugars to 5-HMF. To our knowledge, there are no reports about using a green catalyst to increase the furanic and phenolic compounds to economic interest using vinasses waste. Using a green catalyst reduces the environmental impacts and the costs generated by the compounds commonly used in their recovery. 

## 2. Results and Discussion

### 2.1. Vinasses Characterization

The vinasse presented a pH of 2.42, which was lower than that reported by [7] in sugarcane vinasse ranging between 3.25 and 4.67. The low pH is given by the natural presence of acids in the medium, such as acetic, butyric, lactic, and formic acid, compounds avoiding the rupture of sugar molecules in simple monomers [20].

The high COD value (29.80 g /L) indicated a high content of organic compounds such as polysaccharides, lignin, hemicellulose, proteins, melanoidin, and wax remained after distillation and accumulated during all processes [11]. For original vinasse (OV), the high value of COD is similar to that reported by other authors in tequila vinasses, with concentrations ranging between 28 and 45 g/L [21]. Similar data were reported by Hernández et al. [22] in vinasses extracted from sugarcane (30.4 g/L) but lower than that found in cane molasses (95 g/L), wine (50.2 g/L), and sorghum (79.9 g/L). The high content of sugars and phenolic compounds will be discussed later.

### 2.2. COD Removal

After the thermal treatment, the vinasses presented a reduction in their percentage of COD. Treatments at 110 °C presented low COD removal regardless of time or catalyst (*p* < 0.05). Higher temperatures (120 °C and 130 °C) increased the COD removal regardless of the time or catalyst concentration. T130:40m:0.5P treatment had the highest COD removal (32.9%) (Table 1). The temperature, time, catalyst, and interactions significantly affected the COD removal (Table 2). In rice vinasses, the COD removal could be 83% using a fermentative process [23], and in wine vinasse, the reduction could be 40 to 60% with an anaerobic reactor [24].

### 2.3. Furans Increase

The presence of furanic compounds such as 5-HMF and furfural has been previously reported in tequila vinasses obtained by artisanal processes [25]. Rodríguez-Félix et al. [13] identified 11 furan compounds, including furfural (52.1 mg/L) and 5-HMF (347.6 mg/L) in tequila vinasses, from two processes of production. Rodríguez-Romero et al. [26] analyzed different tequila vinasses and found concentrations of 5-HMF from 18.8 to 227 mg/L. The variation of the 5- HMF and furfural concentrations depends on the samples’ distillation process and storage time [13]. 

After thermal treatments with Taguchi experimental design, it was observed that at 130 °C, 40 min, and 0.5% of catalyst, the 5-HMF concentration significantly increased (127 mg/L) compared to other OV, and it was the better treatment to improve the 5-HMF formation (Figure 1A–C).

Other thermal treatments in combinations with different times and catalysts significantly increase the 5-HMF concentration from 3- to 8-fold compared to OV (Table 1). However, 120 °C with 30 and 40 min or 130 °C with 20 min treatments had no significant differences. The sugar remaining from OV could be transformed to 5-HMF by isomerization and dehydration through thermal treatments and catalysts (H_2_O_2_) [27] (Figure 2A).

The treatments at 130 °C showed highly significant differences, indicating that time and temperature interfere with the increase of 5-HMF (Table 1). Jeong [28] obtained 5-HMF from the green microalgae *Chlorella* sp. using the thermal treatment and ferric sulfate as a catalyst. This treatment increased 37.2% of 5-HMF at 170 °C for 60 min.

The original vinasse had an initial concentration of furfural of 0.08 mg/L. At low temperatures (110 °C) and short time (20 and 30 min), furfural concentration was not detected. In contrast, at 120 and 130 °C and 30 or 40 min, the furfural concentration significantly increased up to 38-fold with 3.07 mg/L (T130:40m:0.5P) compared to OV (Table 1).

Rodríguez-Félix et al. [13] reported 52.11 mg/L of furfural in tequila vinasses. It is possible to find hemicellulose in the OV derivate of the mill agave process. It could be transformed to xylose and after furfural by dehydration at high temperatures [16] (Figure 2B). Wang et al. [29] reported a yield of 76.6% of furfural from commercial xylose and using mesoporous carbon derived from sulfonated lignin, initiating the 100% conversion of xylose to hydrolysis at 200 °C for 45 min.

The treatments at 120 and 130 °C had the highest recovery of furans (5-HMF and furfural). Nevertheless, the interaction of three factors (temperature, time, and catalyst) is necessary to promote an increase in the concentration of these compounds (Table 2).

The use of green catalysts arises as an alternative for substituting homogeneous catalysts such as organic acids, mineral acids, metal chlorides, metal triflates, and ionic liquids, or heterogenic catalysts such as resins, acidic zeolites, metal oxides, acidulated metal oxides, metal phosphates, silicas and clays functionalized, carbonaceous acids, and magnetic materials [30]. Sweygers et al. [31] obtained 5-HMF and furfural using HCl as a catalyst and microwave-assisted in bamboo (*Phyllostachys nigra* “Boryana”), they had a yield of 21% 5-HMF and 13% of furfural. However, different authors report that homogeneous catalysts such as HCl and H_2_SO_4_ are used most frequently as they are efficient and low cost. However, drawbacks of these catalysts include the highest corrosion in waste generation and separation problems [32].

### 2.4. Sugar Removal

For total sugars, the tequila vinasses presented a concentration of 13.6 g/L, such as the values reported in tequila vinasses (8 to 18 g/L), obtained from different factories [20,21].

After the thermal treatment, the vinasses reduced their total sugar content (31–57%) (Table 1). This reduction is attributed to converting from reductors sugars and lignocellulosic compounds to 5-HMF and furfural [16].

Treatments T130:40m:0.5P and T130:20m:1P had the highest removal percentages, with 57.3 and 55.1%, respectively. However, a significantly lower total sugar removal was observed in the T130:30m:0P treatment than in T130:40m:0.5P and T130:20m:1P. This behavior could be due to the absence of hydrogen peroxide as a catalyst, which favors the oxidation of organic matter and promotes the decomposition of polysaccharides into monomers to form furanic compounds [33]. The temperature and catalyst temperature interactions significantly affected total sugar removal (Table 2).

Total sugar from OV, had sucrose (30.9 mg/L), glucose (100.1 mg/L), fructose (26.8 mg/L), and xylose (18.5 mg/L). Santos et al. [34] reported that sugarcane vinasse contains 3.27 g/L of glucose and 4.4 g/L of fructose. Values higher than those obtained in this work.

The thermal treatment significantly decreased sucrose content, with interval removal from 85.3% to 99.9% (Table 1). This removal was more critical for the increase in the temperature and time than the catalyst concentration (Table 2). The highest decrease in sucrose was observed in the T130:40m:0.5P treatment (99.9%), following T120:40m:0P (99.7%) (*p* < 0.05).

Glucose was the sugar found in the highest concentration in the original vinasse (100.1 mg/L). The thermal treatments promoted glucose removal from 68.3 to 84.8% (Table 1). The highest decrease in glucose was observed in the T120:20m:0.5P treatment (84.8%), followed by T110:30m:0.5P (84.1%), both with 0.5 of catalyst. However, all the treatments had no significant differences between them. The temperature, time, and interactions significantly affected glucose removal (Table 2).

In the case of fructose, thermal treatments in combination with different times and catalysts significantly increase the fructose concentration from 0.8- to 3.7-fold compared to OV (26.8 mg/L). The highest increase was observed in T130:40m:0.5P treatment, followed by T110:30m:0.5P up to 99.6 mg/L and 86.4 mg/L (Table 1). Only the interaction of the catalyst with a temperature significantly affected fructose increase (Table 2).

After treatment, it was observed that at 120 °C, 20 min, and 0.5% of catalyst, the xylose concentration increased by ten-fold compared to OV (18.49 mg/L) in T120:20m:0.5P treatment (Table 1). Thermal treatments had no significant differences, reaching high xylose concentrations from 29.4 to 184.5 mg/L. In the three-temperature evaluated, and with different times but without catalyst or low concentration (0.5%), the xylose concentration increased to the highest values without significant differences. However, with an increase of time (T130:40m:0.5P), the xylose concentration had the lowest increase (29.47 mg/L). The temperature significantly affected the xylose concentration (Table 2).

Vinasses are mainly associated with lignocellulosic compounds (cellulose, lignin, and hemicellulose, respectively) [35]. The hemicelluloses can be readily hydrolyzed into xylose through interaction with organic acids, alkaline solutions, and organic solvents [36].

The increase in fructose could be related to the isomerization of glucose to fructose and the thermal capacity for the rupture and oxidation of sucrose, which generates fructose [35]. It could be by two possible mechanisms. The first would be by thermal hydrolysis of cellulose, leaving free glucose molecules, and the second by thermal hydrolysis of sucrose, leaving free fructose and glucose. The glucose could be transformed into fructose through isomerization mechanisms, followed by conversion to 5-HMF by dehydration (Figure 2A) [35].

Hemicellulose would be readily hydrolyzed to xylose by thermal hydrolysis and can be transformed into value-added products [34,36,37]. Researchers such as Li et al. [38] and Wang et al. [39] found that the xylose applying temperatures higher than 100 °C could form furfural, retro aldol, and formic acid. Figure 2B shows the reaction proposal in the formation of furfural from hemicelluloses presented in the samples.

### 2.5. FTIR from Tequila Vinasses

The FTIR of OV was compared with the treatment that presented the highest yield in the formation of furans and phenols (T130: 40m:0.5P) (Figure 3). The spectrum showed a band at 3300 and a peak at 1334 cm^−1^. These are characteristic of OH bonds, assigned to the stretching and torsion vibrations of the hydroxyl groups of 5-HMF [40]. In the T130:40m:0.5P spectrum, the peak signal appeared more intense (3300 and a peak at 1334 cm^−1^). It could be due to the increase of 5-HMF in the sample.

The signals at 2980 and 2900 cm^−1^ are associated with aliphatic alkyl groups and aromatic rings (CH_3_ and CH_2_) in lignin [41]. Another peak occurs at 1647 cm^−1^, a characteristic peak of the C=O double bond in lignin and phenolic compounds. This peak represents the absorption capacity of lignin. The intensity of the peak indicates that the sample had thermal treatment, indicating changes and ruptures in the structure of the compound [42]. The increase in the intensity of this signal could suggest the formation of phenolic compounds caused by the depolymerization of lignin due to the thermal treatment and the catalyst used [43] (Figure 2C).

The peaks observed at 1450 cm^−1^ indicated the stretching of the C-C bonds, present in both aromatic and olefinic compounds [44]. In addition, olefinic and aromatic C-H bonds were detected in the spectrum at 1280 cm^−1^, which are characteristic signals of phenolic compounds [45].

Intense peaks were present at 1084, 1042, and 881 cm^−1^; these could be attributed to the stretching of C-O, C-C, and C-H bonds in cellulose, hemicellulose, and polysaccharides [46]. The increase in these peaks could be caused by decomposing the lignocellulosic compounds and sugars in the vinasses because treatments leave free simpler molecules.

Lignocellulosic compounds present a decomposition process at temperatures from 100 to 190 °C. Then they reach their glass transition, their complex crystalline structure loses its rigidity, and they become more amorphous than the original structure. They are more exposed to changes in their composition, mainly in a system of constant temperature and pressure [47].

Some authors reported reaction pathways to recover 5-HMF from unconventional raw materials [47]. However, they did not mention the possible mechanisms of 5-HMF formation from vinasse. The main components that could be intervened in furans formation are cellulose and sucrose. When applying thermal and chemical modification, the structures of the polysaccharides tend to fracture, leaving free monomers such as fructose and glucose, essential components in forming 5-HMF [48].

### 2.6. Phenol Compounds Identification

Caffeic acid was determined at a concentration of 0.002 mg/L in the original vinasse. After treatments such as T110:20m:0P, T110:40m:1P, T130:30m:0P, T130:40m:0.5P, and T120:30m:1P, the caffeic acid significantly increased 35.5, 34, 28.5, 25 and 22-fold, respectively, compared to the original vinasse. However, these treatments were not significantly different (Figure 4).

Ferulic acid presented a significant increase from 3.4- to 3.7-fold in the treatments T130:30m:0P, T130:40m:0.5P, and T110:20m:0P. The ferulic concentration was not detected at 120 °C with different times and catalysts (Figure 4).

The 3-hydroxybenzoic acid presented an increase from 1.04- to 1.43-fold their initial concentration in T12O:30m:1P and T130:40m:0.5P, but it was not significantly different. The treatments with temperatures of 110 and 120 °C and concentration catalysts 0.5 and 1% presented a significant decrease from 16.6 to 26.4% (Figure 4). 

Chlorogenic acid recorded a concentration of 0.014 mg/L in the original vinasse. The treatment with the highest and most significant increase was T130:40m:0.5P, with 25.7-fold more than OV (Figure 4).

In the case of vanillic acid, the concentration increased in all treatments without a significant difference between them. The original vinasse had 0.994 mg/L of vanillic acid, and the highest concentration was observed in the treatment T130:40m:0.5P (2.754 mg/L) (*p* < 0.05) (Figure 4). The temperature, time, temperature-time, catalyst temperature, and catalyst-time interaction significantly affected the vanillic increase (Table 2).

After almost all treatments, syringic acid significantly increased from 1.19- to 2.80-fold compared to OV (0.265 mg/L). The highest concentration was in the treatment T130:30m:0P (0.744 mg/L) (Figure 4). The temperature, time, or catalyst or their interaction does not affect the increase of syringic acid (Table 2).

Sinapic acid was detected only in the treatments T110:20m:0P and T120:30m:1P with a concentration of 0.051 and 0.061 mg/L, respectively, and significantly higher than OV. The compounds were not detected at the highest temperatures of 120 °C and 30 min (Figure 4). 

2,4-di-tert-butylphenol acid significantly decreased from 23.9 to 88.9% compared to its initial concentration (1.743 mg/L). The treatments with temperatures of 130 °C presented the highest decrease (Figure 4). Temperature, temperature-time, and temperature-time-catalyst interactions significantly affected 2,4-di-tert-butylphenol acid concentration (Table 2).

On the other hand, salicylic acid significantly decreased with the treatments and was only detected in T110:40m:1P and T130:20m:1P (Figure 4). Temperature, time, or catalyst did not affect salicylic acid concentration (Table 2).

The coumaric acid was detected in low concentration in OV (0.002 mg/L). After treatments, it was only detected with a significant increase (5.5 to 6.0-folds) in T120:40m:0P, T130:20m:1P, T130:30m:0P, and T130:40m:0.5P treatments (Figure 4). The temperatures significantly affected coumaric acid increase (Table 2). In sugarcane vinasse, concentrations of 19 and 7.8 mg/L have been determined for vanillic and caffeic acid, respectively, higher values than those obtained in this study. While gallic, ferulic, and coumaric acids were not detected [49].

Other authors have reported phenolic compounds from distillation processes from sugarcane, such as chlorogenic (1.52 mg/L), caffeic acid (0.20 mg/L), and coumaric (1.4 mg/L) [50]. In the wastewater from the molasses distillery, was identified hydroxybenzoic (73.77 mg/L), chlorogenic (11.83 mg/L), vanillic (25 mg/L), caffeic (147.7 mg/L), syringic (54.8 mg/L), coumaric (424.6 mg/L) and ferulic acid (255 mg/L). Chatterjee et al. [51] reported phenolic compounds in baijiu vinasses (white liquor). These were such as vanillic (15.13 mg/L), chlorogenic (1.74 mg/L), coumaric (0.97 mg/L), sinapic (2.16 mg/L), caffeic (6.36 mg/L), ferulic (1.67 mg/L) and syringic acid (0.54 mg/L). All determinations presented a high concentration compared with this work.

Phenolic acids have been proposed as a potential treatment for many disorders and diseases [51]. Caffeic acid is proposed for diseases related to oxidative stress [52], and vanillic acid has been associated with various pharmacological activities, specifically in cardiovascular diseases [53]. The 3-hydroxybenzoic and chlorogenic acids are related to treating diabetes and metabolic disorder [54]. For this reason, it the importance to increase these compounds from tequila vinasses.

### 2.7. Analysis of Principal Components

Figure 5 shows the principal components analysis (PCA). The PCA indicated that principal component 1 (PC1) explained 32.18% of data variability, while principal component 2 (PC2) explained 25.29%. Both components explained 57.47% of the data variability. The treatments T120:30m:1P, T130:30m:0P, and T120:40m:0P were found in the positive axis of PC1 and were correlated with the highest concentrations of phenolic compounds as syringic, vanillic, and caffeic acid. In addition, the temperature and catalyst interactions significantly affected fructose increase (Table 2).

On the other hand, the total sugar removal, COD removal, glucose, sucrose, pH, and phenolic compounds as salicylic, sinapic, and 2,4 di-tert-butylphenol acid were found in the negative axis of the PC1 component. These compounds decreased when thermal treatments were applied and negatively correlated with treatments T120:30m:1P, T130:30m:0P, and T120:40m:0P.

In the negative PC2, the T110:20m:0P, T110:30m:0.5P, T110:40m:1P, and T120:20m:0.5P treatment was correlated positively with a high amount of cumaric acid and negatively correlated with the treatments found in the positive side from PC2. The T130:20m:1P and T130:40m:0.5P were correlated positively with an increase in the furan’s concentration (5-HMF, furfural) and phenols compounds as acid 3-hydroxybenzoic, chlorogenic, and ferulic acid on PC2 positive axis.

## 3. Materials and Methods

### 3.1. Original Vinasse (OV)

The vinasses are the wastewater obtained after tequila distillation which is done after the fermentation process of agave sugar through yeasts [55]. Samples were collected from tequila factories with an artisanal process (with cooking) located in Arenal, Municipality of Jalisco, Mexico. The sample was collected from the storage tank and mixed at environmental temperature immediately after vinasses were obtained at the final tequila process. After transportation, the vinasse was kept at 4 °C. The original vinasse was characterized by pH 2.42, COD 29.80 g/L, and total sugars 13.61 g/L.

### 3.2. Experimental Design

The optimal variables for the recovery of furans and phenolic compounds were determined in an experimental design Taguchi L9 array (3^3^) with three factors (temperature, time, and catalyst) and nine runs. The number and variables of each experiment are presented in Table 3. Three-dimensional response surface plots obtained from the predictive models Taguchi (Minitab 20.3, State College, PA, USA) were used to illustrate the main and interactive effects of the independent variables on the response variables.

### 3.3. COD and Total Sugars

The Chemical Oxygen Demand (COD) was determined using a digester test (Hatch, Mississauga, ON, Canada). COD digestion vials (20–1500 mg/L) and a DR 5000™ UV-Vis Spectrophotometer (Hatch, Mississauga, ON, Canada) were used. The digester was preheated to 150 °C. Meanwhile, vinasses samples were prepared at a 1:100 dilution, and 2 mL of vinasses was added to the vials. They are stirred and placed in the digester at 150 °C for 2 h. Subsequently, they were allowed to cool down to 10 °C and analyzed into the spectrophotometer. All samples were in triplicate.

Reagent-grade glucose (C_6_H_12_O_6_), sulfuric acid (H_2_SO_4_), and phenol (C_6_H_5_OH) at 5% (*w*/*v*) were used to determine the total sugar in vinasses. The glucose curve ranged from 20 mg/L to 100 mg/L. The samples were prepared in glass vials using 1 mL of vinasse, 600 µL of phenol, and 3.5 mL of concentrated sulfuric acid (97% purity). The vials were allowed to cool to room temperature. Samples are measured on the DR 5000™ UV-Vis Spectrophotometer (Hach, Mississauga, ON, Canada) at 490 nm [56].

### 3.4. Furans Determination

The furans were quantified in a UPLC-MS (XEVO TQS Micro) (Waters, Milford, CT, USA) integrated by an ACQUITY Class I Waters (Milford, CT, USA) binary pumping system. The system was equipped with a waters brand BEH C_18_ column 1000 mm × 2.1 mm × 1.7 µm (Waters, Dublin, Ireland) and probe type APcI (Table 4). The temperature was 150 °C, the probe temperature was 450 °C, the cone gas flow was 100 L/h, and the desolvation gas flow was 600 L/h. The acquisition was performed in multiple reaction monitoring (MRM) modes. The mobile phases: (A) water: formic acid (99.7:0.3 *v/v*) and (B) methanol. The elution gradient was as follows: 70% A as an initial condition, 70% A at 1.5 min, 50% A at 2 min, 0.1% A at 2.01 min, 0.1% A at 3 min, 70% A at 3.01 min and 70% A at 5 min. The quantification of the compounds was carried out by preparing calibration curves with the standards 1, 2.5, 5, 12.5, 25, and 50 mg/L for furans.

### 3.5. Phenolic Compounds Determination

The phenol compounds were quantified in a UPLC-MS (XEVO TQS Micro) (Waters, Milford, CT, USA) integrated by an ACQUITY Class I Waters (Milford, CT, USA) binary pumping system. The system was equipped with a waters brand BEH C_18_ column 1000 mm × 2.1 mm × 1.7 µm (Waters, Dublin, Ireland) and probe ESI (Table 4). The source temperature was 130 °C, the probe temperature was 450 °C, the cone gas flow was 20 L/h, and the desolvation gas was 800 L/h. The acquisition was performed in multiple reaction monitoring (MRM) modes. The gradient with mobile phases: (A) water: formic acid (99.7:0.3 *v/v*) and (B) methanol. The elution gradient was as follows: 90% A as the initial condition, 30% A at 11 min, and 90% A at 15 min. The quantification of the compounds was carried out by preparing calibration curves with the standards 0.1, 0.5, 1, 2.5, and 5 mg/L for phenolic compounds.

### 3.6. Sugar (Glucose, Fructose, Sucrose and Xylose) Determination

The system UPLC-MS (XEVO TQS Micro) was equipped with a Waters brand BEH Amide column 50 mm × 2.1 mm × 1.7 µm (Waters, Dublin, Ireland), and the temperature was maintained at 60 °C to quantify reducing sugars. The source temperature was 120 °C, desolvation temperature 400 °C, cone gas flow 50 L/h, and desolvation gas flow 600 L/h. The acquisition was performed in Single Ion Recording (SIR) mode (Table 5). The mobile phases were A) acetonitrile: water (10:90 *v/v*) + 0.1% ammonia and B) acetonitrile: water (90:10 *v/v*) + 0.1% ammonia. The elution gradient was as follows: 0.1% A as the initial condition, 20% A at 2 min, 60% A at 6 min, 0.1% A at 6.50 min, and 0.1% A at 8 min. The acquisition was performed in single ion recording (SIR) mode. The quantification of the compounds was carried out by preparing calibration curves with the standards 5, 10, 15, 20, 25, and 50 mg/L intervals of the compounds.

### 3.7. FTIR of Tequila Vinasses

The increase of reducing sugars, furans, and phenolic compounds after treatments indicated that the vinasses could have had lignocellulosic compounds in their composition.

Then, the structural characterization of vinasses was determined from Fourier-transform infrared spectra (FT-IR) on a Perkin Elmer Spectrum 100/100 N model (Shelton, CT, USA) in the range of 4000–500 cm^−1^ in the transmittance mode, with a resolution of 16 cm^−1^ and eight scans.

### 3.8. Statistical Analysis

Statistical analyzes were performed with Minitab 20.3 and XLSTAT 2020.2.1 software. The Taguchi analysis was established in the L_9_ matrix. All results obtained were subjected to a Principal Components Analysis (XLSTAT 2020.2.1, USA), Tukey Studentized Rank Analysis (HSD) (Type III), Analysis of variance (ANOVA) (Minitab 20.3, USA), and Tukey with a probability of 5% (*p* < 0.05) with a confidence level of 95% to analyze the differences between the means of the compounds from each treatment.

## 4. Conclusions

Vinasses could be revalued to produce 5-HMF, furfural, and phenols with thermal treatments and a green catalyst such as H_2_O_2_. The recovery of 5-HMF (127.1 mg/L) and furfural (3.07 mg/L) from the oxidative dehydration of hexoses with the help of H_2_O_2_ and acids naturally present in the vinasse represents an environmentally friendly recovery method. In the case of phenols, thermal treatments allowed the decomposition of lignin, increasing the presence of mainly caffeic and vanillic acid. The best treatment was 130:40m:0.5P in terms of higher COD and total sugar removal and increased 5-HMF, furfural, chlorogenic, and vanillic acid.

## Figures and Tables

**Figure 1 molecules-28-01063-f001:**
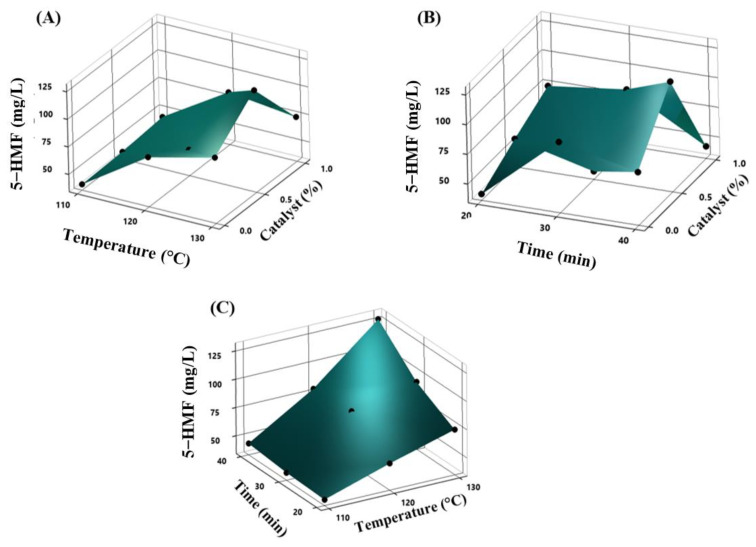
Response surface plots showing the increase in 5-HMF with (**A**) the effects of temperature and time, (**B**) the action of temperature and catalyst, and (**C**) the interaction of time and the catalyst after the treatments.

**Figure 2 molecules-28-01063-f002:**
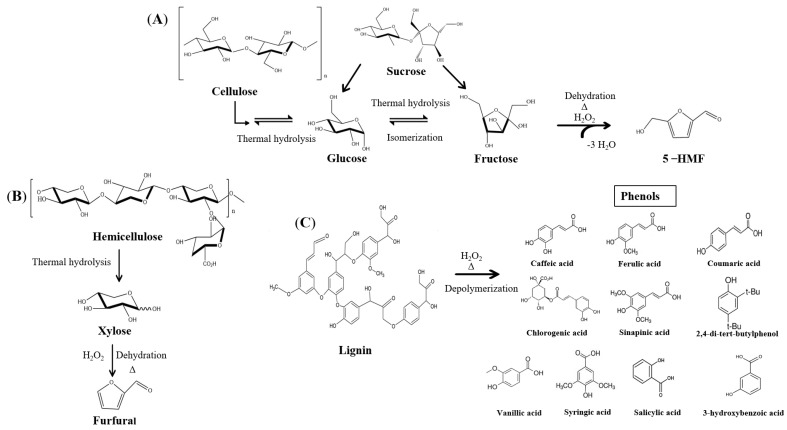
Expected reaction formulas of (**A**) thermal hydrolysis of cellulose and sucrose in the formation of 5-HMF, (**B**) thermal hydrolysis of hemicellulose in the formation of furfural, and (**C**) depolymerization of lignin in the production of phenols, using H_2_O_2_ as a catalyst.

**Figure 3 molecules-28-01063-f003:**
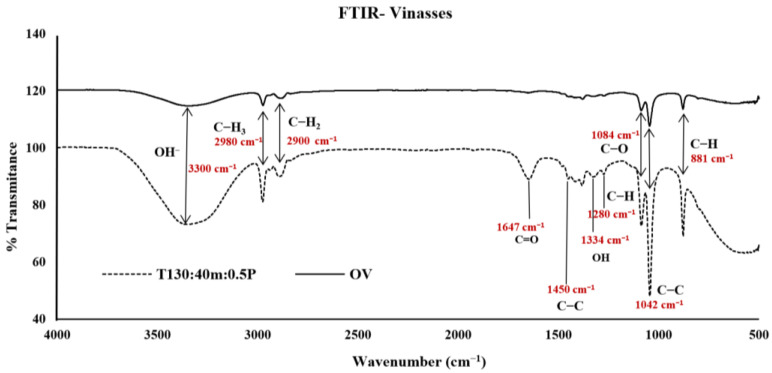
FTIR spectrum of original vinasse (OV) and treatment T130:40m:0.5P.

**Figure 4 molecules-28-01063-f004:**
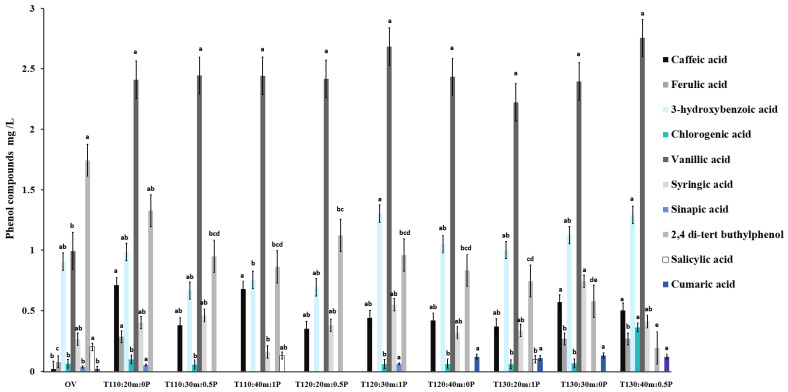
Phenolic compounds were identified in the original vinasses (OV) and after the treatments. Different lower letters between columns are not significantly different for each compound.

**Figure 5 molecules-28-01063-f005:**
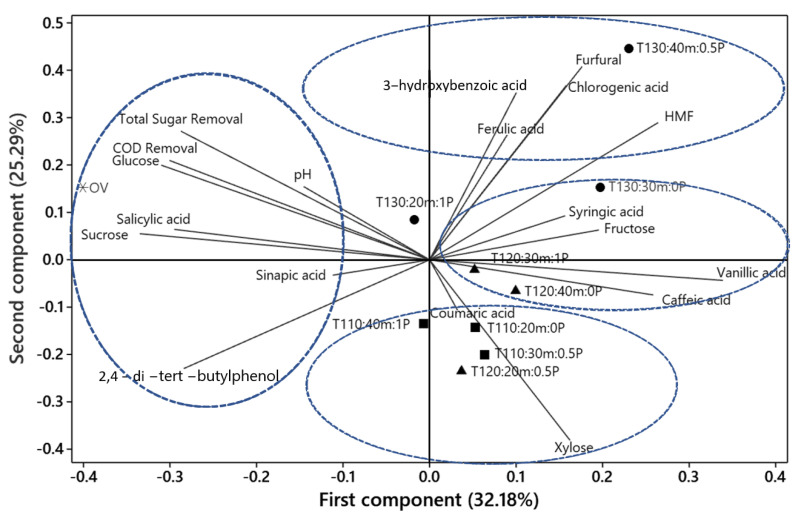
Principal component analysis of furans and phenols present in the samples at the end of the treatments.

**Table 1 molecules-28-01063-t001:** Analysis of the elimination of COD and total sugars, formation of 5-HMF, furfural, and consumption of reducing sugars after thermal treatments.

Treatment	COD Removal (%)	Total Sugars Removal (%)	HMF mg/L	Furfural mg/L	Sucrose mg/L	Fructose mg/L	Xylose mg/L	Glucose mg/L
OV	-	-	10.45 ± 0.01 ^g^	0.08 ± 0.01 ^d^	30.95 ± 0.02 ^a^	26.80 ± 0.04 ^cd^	18.49 ± 0.04 ^c^	100.1 ± 0.10 ^a^
T110:20m:0P	14.1 ± 2.0 ^c^	31.1 ± 3.7 ^c^	39.41 ± 1.2 ^f^	n.d.	5.1 ± 0.2 ^b^	62.8 ± 0.2 ^abc^	161.4 ± 0.2 ^ab^	23.9 ±0.107 ^b^
T110:30m:0.5P	16.2 ± 4.4 ^c^	34.2 ± 3.4 ^bc^	40.84 ± 0.7 ^f^	n.d.	4.4 ± 0.3 ^bc^	86.4 ±0.07 ^ab^	177.8 ± 0.04 ^a^	15.9 ± 0.019 ^b^
T110:40m:1P	14.1 ± 0.9 ^c^	36.5 ± 4.3 ^bc^	44.23 ± 1.3 ^e^	0.17 ± 10.1 ^c^	2.8 ± 0.1 ^cd^	77.9 ± 0.1 ^ab^	95.0 ± 0.7 ^abc^	16.21 ± 0.059 ^b^
T120:20m:0.5P	24.9 ± 4.5 ^b^	31.1 ± 4.4 ^c^	57.66 ± 1.35 ^d^	n.d.	2.0 ± 0.5 ^de^	67.3 ± 0.05 ^ab^	184.5 ±0.246 ^a^	15.2 ± 0.053 ^b^
T120:30m:1P	28.1 ±0.7 ^ab^	42.6 ± 0.4 ^b^	81.73 ± 1.45 ^c^	0.88 ± 1.3 ^b^	0.4 ± 0.1 ^ef^	22.8 ± 0.03 ^d^	159.5 ± 0.430 ^ab^	21.7 ±0.008 ^b^
T120:40m:0P	29.3 ± 0.5 ^ab^	34.9 ± 4.0 ^bc^	79.11 ± 1.60 ^c^	1.33 ± 0.5 ^ab^	0.1 ± 0.1 ^f^	48.6 ± 0.2 ^bcd^	177.0 ± 0.357 ^a^	17.7 ± 0.023 ^b^
T130:20m:1P	28.6 ±1.1 ^ab^	55.1 ± 0.9 ^a^	74.16 ± 2.96 ^c^	0.90 ± 2.6 ^b^	0.2 ± 0.2 ^f^	75.3 ± 0.1 ^ab^	71.5 ± 0.445 ^bc^	21.1 ± 0.062 ^b^
T130:30m:0P	31.1 ± 4.5 ^ab^	34.6 ± 1.6 ^bc^	93.80 ± 2.49 ^b^	1.68 ± 0.4 ^ab^	0.2± 0.04 ^f^	64.4 ± 0.2 ^ab^	125.5 ± 0.197 ^ab^	24.5 ± 0.032 ^b^
T130:40m:0.5P	32.9 ± 2.4 ^a^	57.3 ± 2.7 ^a^	127.0 ± 9.71 ^a^	3.07 ± 0.3 ^a^	0.02 ± 0.01 ^f^	99.7 ± 0.03 ^a^	29.47 ± 0.117 ^c^	31.7 ± 0.018 ^b^

Values shown after ± correspond to the standard deviation; n.d.: not detected. OV: Original vinasse; Different lower letters between columns are not significantly different for each compound.

**Table 2 molecules-28-01063-t002:** The effect of temperature, time, and catalysts on the different compounds.

Compounds	Temperature	Time	Catalyst	Temperature × Time	Catalyst × Temperature	Catalyst × Time	Temperature × Time × Catalyst
COD	<0.0001	0.156	0.759	<0.0001	0.002	0.508	0.017
Total sugars	0.012	0.071	0.224	0.090	0.033	0.204	0.096
5-HMF	<0.0001	<0.0001	0.556	<0.0001	<0.0001	0.010	0.002
Furfural	0.349	0.475	0.309	0.263	0.431	0.305	0.005
Fructose	0.086	0.415	0.189	0.240	0.048	0.435	0.085
Glucose	<0.0001	<0.0001	0.490	0.002	0.002	0.006	0.063
Sucrose	<0.0001	<0.0001	0.525	<0.0001	<0.0001	0.005	0.001
Xylose	0.037	0.297	0.956	0.058	0.119	0.570	0.127
Caffeic acid	0.605	0.550	0.311	0.639	0.608	0.562	0.679
ferulic acid	0.477	0.999	0.312	0.843	0.347	0.821	0.767
3-hidroxybenzoic acid	0.325	0.863	0.760	0.581	0.586	0.945	0.825
Chlorogenic acid	0.482	0.706	0.514	0.653	0.642	0.809	0.809
Vanillic acid	0.002	0.001	0.426	0.015	0.025	0.012	0.161
Syringic acid	0.606	0.068	0.817	0.108	0.780	0.147	0.153
Sinapic acid	0.663	0.663	0.516	0.810	0.810	0.810	0.922
2,4-di-tert-buthylphenol acid	0.018	0.082	0.542	0.003	0.120	0.327	0.003
Salicylic acid	0.094	0.094	0.481	0.293	0.062	0.062	0.206
Cumaric acid	0.050	0.828	0.799	0.137	0.109	0.909	0.237

**Table 3 molecules-28-01063-t003:** Proposal Taguchi design for the increase of furans and phenolic compounds.

Treatment Code	Temperature (°C)	Time (min)	Catalyst H_2_O_2_ (%)
T110:20m:0P	110	20	0
T110:30m:0.5P	110	30	0.5
T110:40m:1P	110	40	1.0
T120:20m:0.5P	120	20	0.5
T120:30m:1P	120	30	1.0
T120:40m:0P	120	40	0
T130:20m:1P	130	20	1.0
T130:30m:0P	130	30	0
T130:40m:0.5P	130	40	0.5

**Table 4 molecules-28-01063-t004:** Mass spectrometric conditions for furans and phenolic compounds.

Phenolic Compounds	t_R_ (min)	Ion Mode	MRM (*m/z*)	Cone (V)	Collision (ev)
Caffeic acid	8	-	179 > 135	20	10
ferulic acid	13.9	-	193 > 134	30	15
3-hydroxybenzoic acid	8	-	137 > 93	10	10
Chlorogenic acid	6.5	-	353 > 191	20	15
Vanillic acid	13	-	167 > 108	15	15
Syringic acid	13.9	-	197 > 123	15	22
Sinapic acid	13.9	-	223 > 164	15	15
2,4-di-tert-buthylphenol acid	14	+	207 > 207	10	0
Salicylic acid	14	-	137 > 65	25	18
Cumaric acid	13.9	-	163 > 119	20	10
**Furans**	**t_R_ (min)**	**Ion Mode**	**MRM (*m/z*)**	**Cone (V)**	**Collision (ev)**
Furfural	1	+	97 > 41	25	15
5-HMF	4.4	+	127 > 81	20	20

**Table 5 molecules-28-01063-t005:** Mass spectrometric conditions for reducing sugars.

Reducing Sugars	t_R_ (min)	Ion Mode	SIR (*m/z*)	Cone (V)	Collision (ev)
Xylose	8	-	149 > 59	20	15
Glucose	7	-	179 > 89	10	12
Fructose	7	-	341 > 179	22	5
Sucrose	7	-	341 > 179	22	5

## Data Availability

Not applicable.

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
