# Peer review of "Thermal Treatment to Obtain 5-Hydroxymethyl Furfural (5-HMF), Furfural and Phenolic Compounds from Vinasse Waste from Agave"

_molecules, 2023, doi:10.3390/molecules28031063_

Round 1
Reviewer 1 Report
The experimental article "Thermal treatment to obtain 5-hydroxymethyl furfural, furfural and phenolic compounds from vinasse waste" is devoted to the processing of stillage tequila into 5-hydroxymethylfurfural, furfural, and phenolic compounds using thermal hydrolysis with hydrogen peroxide as a catalyst. The presented topic is very relevant if we consider several types of stillage, but in this case, the bard is described as a waste from the processing of agave into alcohol. The authors proposed thermal hydrolysis, during which sugar residues, cellulose residues, hemicellulose residues, and even lignin residues are converted into useful products. there is a statistical design, according to which the authors found the conditions for achieving the maximum yield of target products. This article corresponds to the publication, since all substances are named according to chemical nomenclature, and the story goes precisely with the participation of various molecules and their transformations. It should be noted that the authors presented a scheme with transformation reactions, demonstrating the complexity and diversity of processes. This kind of honesty of the authors decorates their article. But there are questions about the text and specific comments that, in the opinion of the reviewer, will quickly correct bottlenecks in the material.
Questions and remarks:
1. In the title of the article, it makes sense to indicate the source of the bard "agave". Barda defines the process: obtaining alcohol. And the peculiarity of the source of alcohol allows the article to become very original.
2. Correct a typo in Figure 2A: in the upper left corner, instead of "isomerization" of cellulose, there should be thermal hydrolysis.
3. It is necessary to provide data on how the target products can be isolated from the resulting mixture. It is clear that this is not the subject of the article, but this issue cannot be bypassed.
4. It is necessary to justify the choice of the green catalyst in this study.
Author Response
"Please see the attachment."

Reviewer 2 Report
The manuscript presented a very promising investigation on HMF obtention but have several shortcomings. The main issue is that the discussion is merely a description of the result, with few mentions of other interesting results and lees comparison with other methods.
It is relevant that individual determination of sugar, furans and phenolics is performed, it would be interesting a more detail correlation between the increase of this compound and the treatment conditions.
Introduction should be revised and discussion improve, reference should include more recent publication. English editing is needed.
Questions?
· The authors use a composite sample? Please explain further.
· Why the authors selected Taguchi approach? Instead of Box Behnken or other ?
General Comments
· Section 2.1 and 2.2 have the same title.
· Improve Figure 2
Other Comments
Line 23 – Please indicate the condition of the treatment not by the code.
Line 24/25 – Please clarify if the manuscript is refering to chlorogenic acid and vanilic acid.
Line 52/56 – use of italics on scientific name
Line 59/62 – That information is presented before, please avoid reiteration
Line 86 – what is the meaning of OV.
Line 86 – COD data is presented before, avoid reiteration
Line 96/98 _ please improve Table citation
Line 345 – please expand the COD method
Line 358 -Table 2?
Line 360/370/380 – the method was isocratic mode?
Line 385 – Table 3 is presented after Table 4
Author Response
"Please see the attachment."

Round 2
Reviewer 2 Report
I appreciate your response.